# Impact of CFTR Modulators on Beta-Cell Function in Children and Young Adults with Cystic Fibrosis

**DOI:** 10.3390/jcm11144149

**Published:** 2022-07-17

**Authors:** Claudia Piona, Enza Mozzillo, Antonella Tosco, Sonia Volpi, Francesco Maria Rosanio, Chiara Cimbalo, Adriana Franzese, Valeria Raia, Chiara Zusi, Federica Emiliani, Maria Linda Boselli, Maddalena Trombetta, Riccardo Crocina Bonadonna, Marco Cipolli, Claudio Maffeis

**Affiliations:** 1Department of Surgery, Dentistry, Pediatrics and Gynecology, Section of Pediatric Diabetes and Metabolism, University and Azienda Ospedaliera Universitaria Integrata of Verona, 37126 Verona, Italy; claudia.piona@univr.it (C.P.); chiara.zusi@univr.it (C.Z.); federicaemiliani1@gmail.com (F.E.); claudio.maffeis@univr.it (C.M.); 2Regional Pediatric Diabetes Center, Department of Translational Medical Sciences, Section of Pediatrics, Federico II University of Naples, 80131 Naples, Italy; mozzilloenza@gmail.com (E.M.); francescomr@hotmail.it (F.M.R.); franzese@unina.it (A.F.); 3Regional Pediatric Cystic Fibrosis Center, Department of Translational Medical Sciences, Section of Pediatrics, Federico II University of Naples, 80131 Naples, Italy; chiarettacim@gmail.com (C.C.); valeria.raia@unina.it (V.R.); 4Cystic Fibrosis Unit, Regional Center for Cystic Fibrosis, University and Azienda Ospedaliera Universitaria Integrata of Verona, 37126 Verona, Italy; sonia.volpi@aovr.veneto.it (S.V.); marco.cipolli@aovr.veneto.it (M.C.); 5Department of Medicine, Section of Endocrinology, University and Azienda Ospedaliera Universitaria Integrata of Verona, 37126 Verona, Italy; marialinda.boselli@univr.it (M.L.B.); maddalena.trombetta@univr.it (M.T.); 6Department of Medicine and Surgery, University of Parma, 43121 Parma, Italy; riccardo.bonadonna@unipr.it; 7Division of Endocrinology and Metabolic Diseases, Azienda Ospedaliero-Universitaria di Parma, 43125 Parma, Italy

**Keywords:** cystic fibrosis, cftr modulators, lumacaftor/ivacaftor, elexacaftor-ivacaftor-tezacaftor, oral glucose tolerance test, glucose metabolism, β-cell function, insulin clearance, insulin sensitivity

## Abstract

Background: To date, no consistent data are available on the possible impact of CFTR modulators on glucose metabolism. The aim of this study was to test the hypothesis that treatment with CFTR modulators is associated with an improvement in the key direct determinants of glucose regulation in children and young adults affected by Cystic Fibrosis (CF). Methods: In this study, 21 CF patients aged 10–25 underwent oral glucose tolerance test (OGTT) before and after 12–18 months of treatment with Lumacaftor/Ivacaftor or Elexacaftor-Ivacaftor-Tezacaftor. β-cell function (i.e., first and second phase of insulin secretion measured as derivative and proportional control, respectively) and insulin clearance were estimated by OGTT mathematical modelling. Insulin sensitivity was estimated by the Oral Glucose Sensitivity Index (OGIS). The dynamic interplay between β-cell function, insulin clearance and insulin sensitivity was analysed by vector plots of glucose-stimulated insulin bioavailability vs. insulin sensitivity. Results: No changes in glucose tolerance occurred after either treatment, whereas a significant improvement in pulmonary function and chronic bacterial infection was observed. Beta cell function and insulin clearance did not change in both treatment groups. Insulin sensitivity worsened in the Lumacaftor/Ivacaftor group. The analysis of vector plots confirmed that glucose regulation was stable in both groups. Conclusions: Treatment of CF patients with CFTR modulators does not significantly ameliorate glucose homeostasis and/or any of its direct determinants.

## 1. Introduction

Cystic fibrosis related diabetes (CFRD) affects at least half of adult CF patients [1]. CFRD and earlier glucose tolerance alterations, associated with an increased risk of developing diabetes, are also common at all ages, including children and toddlers [2,3]. Many reports have demonstrated that the primary cause of these alterations is beta-cell dysfunction with deficient insulin secretion [4,5]. The degree of the beta cell defect in CF patients hallmarks each glucose tolerance stage [6]. A reduced insulin secretion leads to negative protein balance and to a catabolic state, which contributes to the deterioration of nutritional status and pulmonary function [7]. Therefore, an increased risk of morbidity and mortality is found in patients with CFRD [8].

In recent years, the introduction of CFTR modulators has provided a novel therapeutic approach for the treatment of CF by correcting the basic defects in the CFTR channel function. Several beneficial effects on the clinical course of CF and its complications have been described [9]. Currently, four main CFTR modulators for people with certain CFTR mutations are available: Ivacaftor, Lumacaftor-Ivacaftor, Tezacaftor-Ivacaftor and Elexacaftor-Tezacaftor-Ivacaftor. According to current studies, Elexacaftor-Ivacaftor-Tezacaftor therapy is more effective in restoring CFTR function and thus in recovering lung function and nutritional status than Lumacaftor/Ivacaftor therapy [10,11].

To date, studies regarding cystic fibrosis transmembrane conductance regulator protein (CFTR) channel expression in pancreatic beta-cell showed inconsistent results. Some studies demonstrated that cystic fibrosis transmembrane conductance regulator protein (CFTR) channel is expressed in pancreatic beta-cells [12,13] and alterations of its function have a negative impact on beta-cell function [14,15]. On the contrary, two studies evaluating CFTR expression in human islet cells found only minimal CFTR mRNA expression and no detectable CFTR protein [16,17].

Another recent study showed heterogeneous expression of CFTR in insulin-secreting β-cells of the normal human islet [18].

To date, some studies have investigated the possible impact of CFTR modulators on glucose metabolism. Regarding Ivacaftor therapy, a five-year prospective registry study first showed a reduction in the CFRD incidence [19], whereas a significant improvement in insulin secretion, in particular its first phase measured during a mixed-meal tolerance test, was observed in young CF patients without diabetes [20]. The few studies in adult patients homozygous for Phe508del mutations treated with Lumacaftor/Ivacaftor showed no consistent results. An amelioration of glucose tolerance abnormalities has been reported after one year of treatment [21], but this finding was not confirmed by the PROSPECT Part B study that reported no significant changes in both glucose tolerance and insulin secretion [22].

These studies evaluated β-cell function, and in particular insulin secretion, by using surrogate markers and not by applying a validated mathematical model based on Oral Glucose Tolerance Test (OGTT) with characterization of the two components of glucose stimulated insulin secretion and of insulin sensitivity and clearance [23,24]. However, a more comprehensive and deep understanding of any change in glucose metabolism requires the simultaneous measurement of at least the three direct determinants of glucose regulation, i.e., β-cell function, insulin clearance and insulin sensitivity [25,26]. β-cell function is evaluated measuring the first and second phase of insulin secretion, whereas insulin clearance and sensitivity allow insulin availability and the responsiveness of cells to insulin action to be assessed, respectively.

Recently, Colombo et al. simultaneously assessed these three determinants in a retrospective case-control study of thirteen Phe508del homozygous patients treated with Lumacaftor/Ivacaftor therapy for one year, and reported that therapy did not improve any glucose regulation mechanism [22,23,24,25,26,27]. Notably, no data are available about the effect of Elexacaftor-Ivacaftor-Tezacaftor combination therapy on the direct determinants of glucose metabolism.

Progress in the knowledge regarding the possible beneficial effect of this therapy on glucose metabolism during the pediatric age could be particularly relevant for preventing the development of CFRD and, thus, its adverse impact on the clinical course of CF.

The aim of this study was to test the hypothesis that treatment with Lumacaftor/Ivacaftor or Elexacaftor-Ivacaftor-Tezacaftor is associated with an improvement of the direct determinants of glucose regulation, i.e., β-cell function, insulin clearance and insulin sensitivity, in a cohort of children and young adults with CF.

## 2. Materials and Methods

### 2.1. Participants and Study Protocol

Twenty-one subjects aged ≥ six years with confirmed CF diagnosis (by positive sweat test and CFTR mutation analysis) in regular follow up at two Regional CF Care Centers (Verona and Napoli) and the two Regional Centers for Pediatric Diabetes of the same University Hospitals for CFRD screening program were enrolled in this prospective observational study. Sixteen subjects, homozygous for Phe508del mutation, underwent treatment with Lumacaftor/Ivacaftor, whereas five patients, with at least one Phe508del mutation with severe lung disease (FEV1 < 40%), underwent treatment with Elexacaftor-Ivacaftor-Tezacaftor for compassionate use.

Exclusion criteria were: changes in antibiotics and/or steroids and/or other medications possibly affecting glucose metabolism in the 6 weeks preceding the study visits with OGTT; clinical history of pulmonary exacerbation and/or symptoms of acute infection in the 6 weeks preceding the study visits with OGTT; severe liver and/or kidney disease; and liver and/or pulmonary transplantation.

Informed consent was obtained from adult patients and the parents/caregivers of paediatric patients. The protocol was approved by the Institutional Ethics Committees of the two participating centres (Verona and Napoli, Italy).

Each participant underwent physical examination, spirometry, sweat chloride test and OGTT one to twelve weeks before starting CFTR modulators therapy and after twelve to eighteen months of treatment.

### 2.2. Clinical Characteristics

A complete physical examination with the collection of anthropometric measurements (body height and body weight) was performed. Body mass index (BMI) was calculated, and BMI values were standardized, calculating age and gender-specific BMI percentiles using WHO child growth standards [28].

Forced Expiratory Volume in the 1st second (FEV1) and Forced Vital Capacity (FVC) were measured by spirometry, and the percentages of predicted (FEV1% and FVC%) were calculated according to current guidelines [29]. A sweat chloride test had been performed according to current international guidelines [30]. Pancreatic insufficiency was defined by the need for pancreatic enzyme replacement therapy and faecal elastase < 200 mg/dL.

The total number of pulmonary exacerbations and the number of pulmonary exacerbations requiring intravenous antibiotic therapy in the 12 months preceding and following CFTR modulators therapy initiation was also recorded.

### 2.3. Oral Glucose Tolerance Test

Standard OGTT (1.75 g/kg, max 75 g) was performed at 08:00 a.m. after overnight fasting. Blood samples for measuring plasma glucose, serum insulin and C-peptide concentrations were taken at baseline and at times +30, +60, +90 and +120 min.

Plasma glucose, insulin and C-peptide levels were analyzed using standard procedures at the central laboratories of the two participating centres. In particular, plasma glucose was measured with the glucose oxidase method. Insulin and C-peptide levels were analyzed by enzyme-immunoassay (Mercodia AB, Sweden).

Both laboratories belong to the Italian National Health System and are certified according to International Standards ISO 9000 (www.iso9000.it/, accessed on 10 May 2022), undergoing semi-annual quality controls and inter-laboratory comparisons.

According to the current guidelines [31], participants were classified in one of the following glucose tolerance classes: (i) normal glucose tolerance (NGT: fasting blood glucose (FPG) ≤ 100 mg/dL (≤6.1 mmol/L), 2-h and mid-OGTT glucose level < 140 mg/dL (<7.7 mmol/L)); (ii) indeterminate glucose tolerance (INDET: FPG ≤ 126 mg/dL (≤7 mmol/L), 2-h glucose < 140 mg/dL (<7.7 mmol/L) but OGTT glucose ≥200 mg/dL (≥11.1 mmol/L) at any mid-time between +30 and +90 min of the test); (iii) impaired glucose tolerance (IGT: 2-h glucose level ≥ 140 (≥7.7 mmol/L) and <200 mg/dL (<11.1 mmol/L)); (iv) diabetes (CFRD: 2-h glucose level ≥ 200 mg/dL (≥11.1 mmol/L)), with and without fasting hyperglycaemia.

In addition, HbA1c value was measured by high-performance liquid chromatography and standardized to the DCCT normal range (4.0–6.0%, 20–42 mmol/mol) on the same day of each OGTT.

### 2.4. Assessment of the Determinants of Glucose Regulation during the OGTT

We used the same modelling strategy previously applied in CF patients [6] and developed in previous publications [10,11].

Beta-cell function is reconstructed as the sum of two components:

(1) Derivative (or dynamic) control (DC): it describes the response of beta-cells to the rate of increase in glucose concentration, i.e., the sensitivity of beta-cells to glucose increase and reflects the first phase of insulin secretion.

(2) Proportional (or static) control (PC): it describes the response of beta-cells to glucose concentration (i.e., the sensitivity of beta-cells to glucose, per se). It can be presented as the insulin secretion rate (ISR) vs. glucose concentration plot. It can also be quantified with the model computed compact parameter σ2 (units: pmol·min^−1^ per mmol·L^−1^), which measures the increment in insulin secretion rate (pmol·min^−1^) in response to the increment of 1 mmol·L^−1^ of glucose concentration, i.e., the slope of the ISR vs. the glucose concentration curve. The PC reflects the second phase of insulin secretion [26].

Insulin clearance (units: L·min^−1^) was computed according to the following formula: ClearanceIns = AUC_ISR_/[AUC_I_ + (I_Final_ − I_Basal_)·MRT_Ins_], in which AUC_ISR_ is the area under the curve of insulin secretion rate (computed by the model), AUC_I_ is the area under the curve of insulin concentration, I_Final_ is the insulin concentration at the end of the OGTT, I_Basal_ is the insulin concentration at time 0′, MRT_Ins_ is the mean residence time of insulin, which was assumed to be 27 min in subjects with diabetes and 18 min in subjects without diabetes, as previously reported [32].

Insulin sensitivity was estimated by the Oral Glucose Sensitivity Index (OGIS) [33]. This index is derived from a mathematical model of glucose metabolism based on established glucose kinetics and insulin action principles.

Insulin sensitivity and glucose-stimulated insulin bioavailability are central elements of the physiological feedback loop which governs glucose homeostasis [26,34].

Glucose stimulated insulin bioavailability is determined by glucose stimulated insulin secretion (i.e., beta-cell function) and insulin clearance. The greater the glucose stimulated insulin secretion, the greater the glucose stimulated insulin bioavailability. The greater the insulin clearance, the lower the glucose-stimulated insulin bioavailability. PC accounts for about 90% of insulin released by the beta-cells in response to OGTT [21]. Thus, glucose-stimulated insulin bioavailability (PC_adj_; units: pmol·mmol^−1^, i.e., pmoles of insulin per mmol of glucose) is computed according to the following formula [26,32]:

PC_adj_ = σ^2^/insulin clearance

Values of PC_adj_ and OGIS, measured before and after the treatment with modulators, were plotted together with the trajectory of the change over time, creating a vector plot, as previously described [26,32]. In this paper, the concave line drawn in the vector plot is the physiological inverse (hyperbolic) relationship between insulin bioavailability and insulin sensitivity found in 11 years of age and gender matched subjects with Phe508del homozygosis, with a normal glucose tolerance test (NGT), selected from the cohort of subjects already analysed in [6]. The area below the concave line houses a less than normal adaptation of glucose-stimulated insulin secretion and insulin clearance to insulin sensitivity. The greater the distance between a point in this area and the concave line is, the worse the body’s adaptation is, and the worse the degree of the alteration in glucose regulation is. An improvement in glucose regulation with return to the normal feedback loop may occur through improved insulin bioavailability (upward vertical vector), improved insulin sensitivity (rightward horizontal vector) or, most commonly, a combination of both (oblique vector) [34].

Several fasting and OGTT-derived biomarkers of insulin sensitivity/resistance and of beta-cell function were also computed [26] according to the following equations, as previously described [6]:-Homeostasis model assessment insulin resistance (HOMA-IR), as marker of insulin resistance based on fasting glycemia and insulin: ((Insulin0′ (mU/L) Glucose0′ (mmol/L))/22.5);-Insulinogenic index (IGI), as marker of early insulin bioavailability in response to oral glucose: (insulin30′ (mU/L)-insulin0′ (mU/L))/(glucose30′ (mg/dL)-glucose0′ (mg/dL));-Matsuda index, as marker of postprandial insulin sensitivity: 10,000/((Glucose 0′ (mg/dL) ·Insulin0′ (mU/L)) · (mean OGTT glucose concentration (mg/dL)) · (mean OGTT insulin concentration (mU/L)))^1/2^;-Oral disposition index (DI), a popular marker of the adequacy of insulin bioavailability to the prevailing insulin sensitivity: Matsuda index·IGI.

### 2.5. Statistical Analysis

The Kolmogorov-Smirnov test was used to assess normal distribution of variables. Patients’ characteristics with normal distribution were reported as mean and standard deviation (SD), whereas not normally distributed variables were reported as median and interquartile range (IQR). Results regarding beta-cell function are plotted as mean and standard error of the mean (SEM) in the figures. Categorical data were presented as absolute frequencies and percent values. Expected PCadj values were calculated from the concave line formula to evaluate the distance from observed PCadj values.

The comparison of clinical and metabolic parameters before and after CFTR modulator’s therapy was performed using paired Student’s *t*-test or Wilcoxon signed-rank, when indicated. A *p* value < 0.05 was considered as statistically significant. Proportional Control before and after CFTR modulators therapy was compared by a generalized linear model for repeated measures, adjusting for age and gender. All the analyses were performed using SPSS v.22.0 (SPSS, Chicago, IL, USA).

## 3. Results

Sixteen subjects, homozygous for Phe508del mutation, were treated with the standard dose of Lumacaftor/Ivacaftor therapy (Lumacaftor 800 mg/Ivacaftor 500 mg daily > 12 years, Lumacaftor 400 mg/Ivacaftor 500 mg daily between 6 and 12 years), whereas five patients, with at least one Phe508del mutation, were treated with Elexacaftor-Ivacaftor-Tezacaftor (morning dose: 100 mg Elexacaftor/50 mg Tezacaftor/75 mg Ivacaftor; evening dose: 150 mg Ivacaftor).

The baseline visit was performed 26.5 days [IQR 21.3 and 32.4] before starting CFTR modulator’s therapy, with a median follow-up period after therapy initiation lasting 16.3 months [IQR 13.9–17.9].

Clinical and metabolic characteristics of study participants before and after 12 to 18 months of treatment with Lumacaftor/Ivacaftor and Elexacaftor-Ivacaftor-Tezacaftor are shown in Table 1 and Table 2, respectively. The comparison of anthropometric parameters, including BMI z-score, measured before and after treatments, showed no significant differences. In contrast, a significant improvement in pulmonary function parameters, number of pulmonary exacerbations and sweat chloride concentration was observed both in subjects treated with Lumacaftor/Ivacaftor and in those treated with Elexacaftor-Ivacaftor-Tezacaftor. The distribution of glucose tolerance stages according to the results of OGTT did not change significantly before and after treatment in both treatment subgroups.

Beta-cell DC and PC measured before starting CFTR modulators therapy and after treatment are presented in Figure 1. In both groups, DC did not change significantly. The derivative control decreased numerically by 18% in subjects treated with Lumacaftor/Ivacaftor (Figure 1, panel a), whereas a numerical increase was detected in patients treated with Elexacaftor-Ivacaftor-Tezacaftor (Figure 1, panel b). The PC, as described by the entire glucose stimulated insulin secretion rate curve over the 4–15 mmol/L range of glucose concentration, was almost superimposable to baseline values after both treatments (Figure 1, panel c and d).

Insulin clearance did not change significantly in either of the two study groups.

Insulin sensitivity, as assessed by OGIS, was significantly lower after Lumacaftor/Ivacaftor with a median decrease of 15.8% [IQR −11.1% to −23.3] (*p* = 0.004) vs. baseline values. OGIS did not change significantly in patients treated with Elexacaftor-Ivacaftor-Tezacaftor.

Figure 2 presents the vector plots of glucose-stimulated insulin bioavailability, as assessed by PC_adj_, vs. insulin sensitivity before and after treatment with Lumacaftor/Ivacaftor (panel a) and Elexacaftor-Ivacaftor-Tezacaftor (panel b). Glucose-stimulated insulin bioavailability was numerically, but not significantly, worse than expected, both at baseline and after treatment. Both vector plots were characterized by a leftward and upward trajectory, with no changes in the distance from the concave curve which represents the NGT status. These patterns represent the absence of changes in glucose regulation after treatment.

Additional metabolic characteristics of patients treated with Lumacaftor/Ivacaftor and Elexacaftor-Ivacaftor-Tezacaftor are shown in Appendix A, respectively.

The comparison of glucose, insulin and C-peptide values measured during the OGTT performed before and after the treatment showed no significant differences, except for fasting plasma glucose, which was significantly higher after Lumacaftor/Ivacaftor (*p* = 0.032). HbA1c level was significantly reduced only in patients treated with Elexacaftor-Ivacaftor-Tezacaftor (*p* = 0.04). The fasting and the OGTT-derived biomarkers of insulin secretion and action did not significantly change in either of the two study groups.

## 4. Conclusions

The main finding of the present study is the evidence that both treatments with Lumacaftor/Ivacaftor and Elexacaftor-Ivacaftor-Tezacaftor have no significant impact on glucose tolerance, on beta-cell function and on the dynamic interplay of the latter with insulin clearance and insulin sensitivity, i.e., on the mechanistic feedback loop which governs glucose regulation.

Treatment with CFTR modulators can enhance or restore the functional expression of specific CF-causing mutations through potentiation, correction, or amplification of CFTR channel function.

In particular, the introduction of Lumacaftor/Ivacaftor and Elexacaftor-Ivacaftor-Tezacaftor represent a revolutionary therapeutic option for patients with at least one F508del mutation in CFTR gene [35]. These two combinations of correctors and potentiators proved to be effective in changing the clinical course of CF, although Elexacaftor-Ivacaftor-Tezacaftor therapy was shown to be more effective in recovering lung function and nutritional status than Lumacaftor/Ivacaftor therapy [10,11]. To date, limited data are available regarding their possible impact on glucose metabolism.

Studies regarding cystic fibrosis transmembrane conductance regulator protein (CFTR) channel expression in pancreatic beta-cell showed inconsistent results. Some studies demonstrated that CFTR dysfunction negatively influence the insulin secretory activities of beta-cells by several direct and indirect pathophysiological processes [12,13]. Moreover, in recent years, increasing evidence has demonstrated that chloride transporters and channels, including CFTR, substantially contribute to modulating β-cell electrical activity and, therefore, insulin secretion [36]. A recent study further supports this evidence, showing that CFTR inhibitors reduce, rather than completely inhibit, the overall secretory response in mouse, rat, and human beta cells, demonstrating that CFTR participates, at least in part, in the secretory response of beta cells, and intrinsic β-cell dysfunction may directly participate in the pathogenesis of CFRD [18].

In particular, intracellular Cl- electrogenically exits through Cl- channels expressed in β-cells results in depolarization of the membrane, because an outwardly directed Cl- gradient is established and then regulated by the balance between Cl- transporters and channels [18].

The lack of Cl- efflux through a mutated channel reduces the glucose-induced membrane depolarization and, consequently, the activation of voltage-gated Ca^2+^ channels.

These alterations result in a reduction in the elevation of intracellular calcium concentrations, which is required for both the first and the second phase of insulin secretion [37]. The first phase consists in the rapid secretion of the readily releasable pool of insulin granules and, according to the model of the glucose-induced triggering pathway, is strongly and directly depolarization-dependent. The second-phase insulin secretion is characterized by a less prominent and slower insulin release from newly mobilized granules mainly driven by voltage-gated Ca^2+^ channels and the metabolic amplifying pathway [38]. According to the pre-clinical evidence, CFTR modulators could potentially influence both these phases.

Currently, published data regarding the possible impact of Lumacaftor/Ivacaftor on glucose metabolism and CFRD are limited. Moreover, previous studies measuring surrogate markers of beta-cell function and insulin sensibility provided conflicting results. Thomassen et al. reported no significant changes in glucose tolerance and insulin secretion, as estimated by AUC of insulin levels measured during OGTT, in five adolescents and adults after six to eight weeks of treatment [39]. These findings were confirmed in a larger cohort of adult patients by the PROSPECT Part B study [22], whereas another study involving a cohort with a comparable sample size showed a clinical improvement in glucose tolerance after one year of treatment [21].

None of these three studies used state-of-the-art tools to measure beta cell function and/or insulin sensitivity, nor assessed comprehensively the direct determinants of glucose regulation. Recently, Colombo et al. evaluated beta cell function, insulin clearance and insulin sensitivity in 13 CF patients homozygous for Phe508del CFTR mutation after 1 year of Lumacaftor/Ivacaftor treatment in comparison to untreated patients with the same genotype [27]. After the treatment, no significant improvements in glucose regulation were found.

To our best knowledge, no data regarding beta cell function, insulin clearance and insulin sensitivity are currently available in patients treated with Elexacaftor-Ivacaftor-Tezacaftor.

In the present study, glucose metabolism before and after treatment with CFTR modulators was investigated through the simultaneous measurement of the direct determinants of glucose regulation, i.e., beta-cell function, insulin clearance and insulin sensitivity, and the evaluation of their dynamic interplay in two cohorts of children and young adults with CF treated with Lumacaftor/Ivacaftor and Elexacaftor-Ivacaftor-Tezacaftor, respectively.

The comparison of DC and PC, which reflect the first and the second phase of insulin secretion, respectively, and insulin clearance revealed no significant changes before and after the treatment with both modulators. These findings agree with those presented by Colombo et al. [27] and also tentatively extend them to the patients treated with Elexacaftor-Ivacaftor-Tezacaftor.

We report a statistically significant decline in insulin sensitivity in patients treated with Lumacaftor/Ivacaftor. However, the vector plots’ analysis (Figure 2), which represents the dynamic interplay between the key direct determinants of glycaemic homeostasis, showed that glucose regulation was stable both in patients treated with Lumacaftor/Ivacaftor and in those treated with Elexacaftor-Ivacaftor-Tezacaftor. Indeed, in both treatments, no worsening/amelioration of the compensatory capacity of beta-cell function was detected.

No significant changes were found in OGTT-derived surrogate indices of insulin secretion and insulin sensitivity, in agreement with the OGTT-derived assessment of beta cell function. Although surrogate indices could be useful in clinical practice, they cannot provide a reliable reconstruction of the architecture of the mechanistic feedback loop of beta cell function, insulin clearance and insulin sensitivity [26], herein reported.

In our study, no substantial changes in glucose tolerance categories, mid-OGTT and 2-h glucose were observed, in agreement with the results of the PROSPECT Part B study conducted in a larger sample of adults treated with Lumacaftor/Ivacaftor therapy for 12 months [22]. A slight increase in fasting glucose was observed in the group treated with Lumacaftor/Ivacaftor therapy, however, it was largely within the normal range in all study participants. Moreover, these results are in line with previous findings, showing that beta-cell (dys)function plays a primary role in determining alterations in the glucose metabolism of CF patients and that its changes hallmark the glucose tolerance stages in CF patients [6].

Notably, we herein report a statistically significant, albeit numerically slight, improvement of HbA1c in patients treated with Elexacaftor-Ivacaftor-Tezacaftor. Interestingly, in a recent study, Scully et al. reported that initiation of Elexacaftor-Ivacaftor-Tezacaftor therapy in 23 adults with CF, was associated with improvement in HbA1c value and several continuous glucose monitoring-derived glycaemic parameters, such as average glucose, standard deviation, or time in range, in patients both with and without CFRD diagnosis [40]. Further studies are needed to confirm these findings and to better explore the reasons for this improvement.

Our study has at least three main limitations: (i) a longer follow up may be needed to detect significant changes in glucose tolerance, glucose stimulated insulin bioavailability and insulin sensitivity; (ii) the sample sizes are small and do not allow specific analysis to be conducted according to glucose tolerance groups; (iii) a limited number of paediatrics subjects were studied.

The main strength of the study is the quality of the evaluation of the dynamic interplay between beta-cell function, insulin clearance and insulin sensitivity, which necessarily underpins any potential change in glucose regulation/homeostasis after treatment with CFTR modulators.

In conclusion, the assessment of the key direct pathophysiologic determinants of glucose regulation in CF patients undergoing CFTR modulators demonstrated that these treatments ameliorated neither glucose-stimulated insulin bioavailability nor insulin sensitivity and, hence, did not improve glucose tolerance.

Further prospective studies are required to corroborate these findings, particularly regarding Elexacaftor-Ivacaftor-Tezacaftor therapy, to longitudinally investigate glucose regulation changes during the longer period of CFTR modulator’s therapy and to identify the specific molecular pathways behind these changes, if any. Moreover, future clinical trials should involve paediatric subjects to evaluate whether an early start of CFTR modulator therapy in childhood can prevent the development of glucose tolerance alterations.

## Figures and Tables

**Figure 1 jcm-11-04149-f001:**
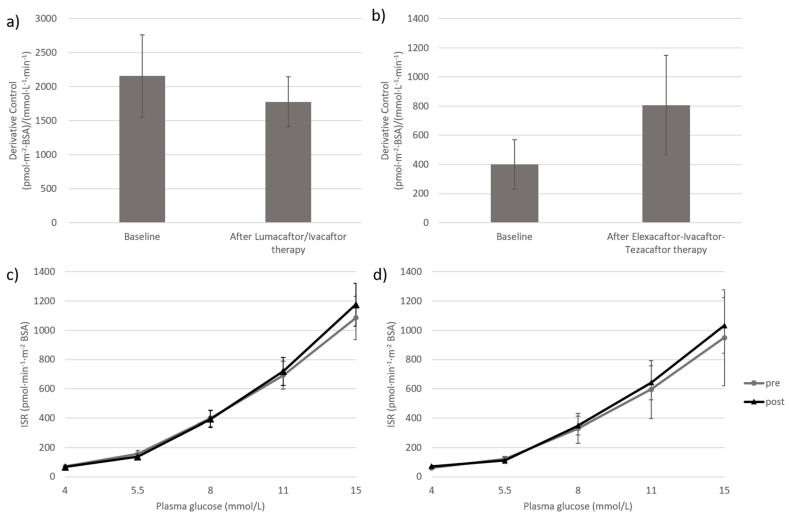
Derivative and Proportional Control before starting CFTR modulators therapy and after 12–18 months of treatments. Effects of Lumacaftor/Ivacaftor therapy and Elexacaftor-Ivacaftor-Tezacaftor on Derivative Control (**a**,**b**) and Proportional Control, i.e., the curve relating insulin secretion rate (*y* axis) to glucose concentration (*x* axis) (**c**,**d**). Differences between groups were tested by Wilcoxon signed rank test (**a**,**b**) and by generalized linear model for repeated measures (**c**,**d**). *p*-value > 0.05.

**Figure 2 jcm-11-04149-f002:**
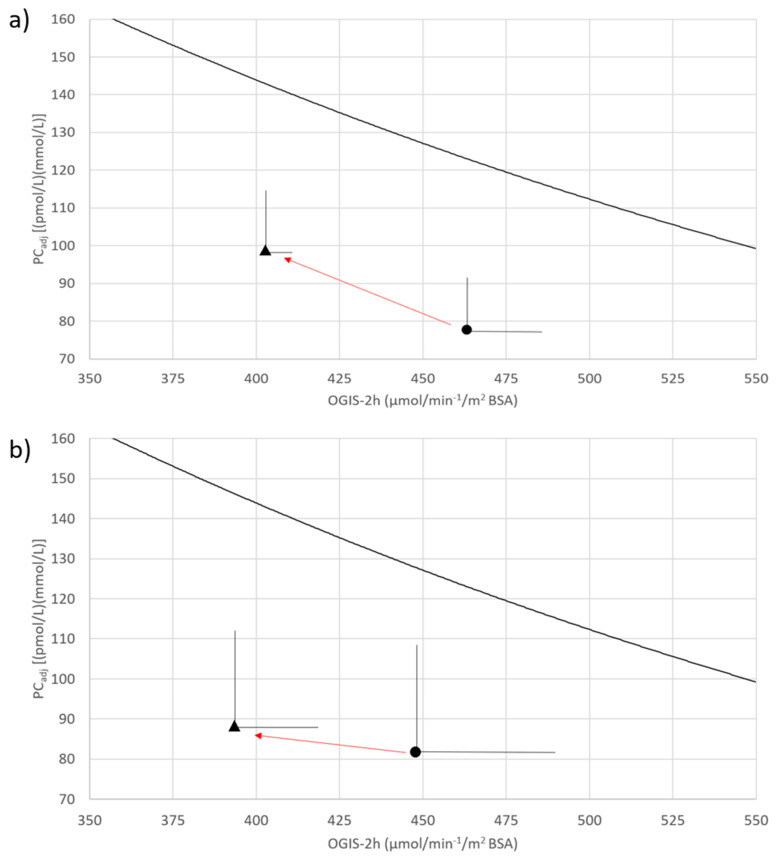
Joint changes in insulin sensitivity (OGIS-2h; *x*-axis) and in glucose stimulated insulin bioavailability (PC_adj_; *y*-axis) before and after CFTR modulators therapy. Points represent the joint action of glucose-stimulated insulin bioavailability (PC_adj_) and insulin sensitivity (OGIS) before (circle) and after (triangle) CFTR modulator’s therapy, while trajectory is the change over time. Lumacaftor/Ivacaftor therapy (**a**) and Elexacaftor-Ivacaftor-Tezacaftor therapy (**b**) effects are respectively represented. The concave line in the vector plots is the physiological inverse (hyperbolic) relationship of the glucose-stimulated insulin bioavailability vs. insulin sensitivity found in 11 individuals, delta508F homozygotic, with normal glucose homeostasis (NGT). The area below the concave line houses the less than normal adaptation to insulin sensitivity. The greater the distance between a point in this area and the concave line, the worse the body’s adaptation, and the worse the glucose regulation. Differences between PC_adj_ values were detected by Students’ *t*-test and by Wilcoxon signed rank test between OGIS-2h values. No significant differences were found except for OGIS-2h before and after Lumacaftor/Ivacaftor therapy (panel **a**, *p*-value = 0.004).

**Table 1 jcm-11-04149-t001:** Clinical and metabolic characteristics of the subjects before starting Lumacaftor/Ivacaftor and after 12–18 months of treatment. Data are expressed as mean ± SD or median [IQR], unless otherwise specified. Abbreviations: BMI body mass index, FEV1 Forced Expiratory volume in the 1st second, FVC forced vital capacity. NGT normal glucose tolerance, INDET indeterminate glucose tolerance, IGT impaired glucose tolerance, CFRD cystic fibrosis related diabetes, DC derivative control, PC proportional control, ISR insulin secretion rate at 4, 5.5, 8, 11 and 15 mmol/L of plasma glucose, OGIS Oral Glucose Insulin Sensitivity.

Variables	Baseline	After 12–18 Months of Lumacaftor/Ivacaftor Treatment	*p*
Gender (M/F) n (%)	9 (56.2)/7 (43.8)	-	
Age (years)	15.5 ± 4.6	17.0 ± 4.5	<0.001
Pubertal status			0.65
Pre-pubertal n (%)	1 (6.3)	1 (6.3)	
Pubertal n (%)	4 (25.0)	2 (12.5)	
Post-pubertal n (%)	11 (68.7)	13 (81.2)	
Height (Z-score)	−0.07 ± 0.70	0.09 ± 0.92	0.16
Weight (Z-score)	−0.47 ± 0.55	−0.40 ± 0.77	0.62
BMI (kg × m^−2^)	18.84 ± 3.20	19.80 ± 3.10	0.06
BMI z-score	−0.63 ± 1.38	−0.27 ± 1.20	0.12
FEV1 (L)	2.50 ± 0.86	2.89 ± 0.90	<0.001
FEV1% of predicted	86.56 ± 16.93	93.56 ± 18.74	0.002
FVC (L)	3.38 ± 1.29	3.95 ± 1.26	<0.001
FVC% of predicted	98.75 ± 24.35	108.59 ± 24.36	<0.001
Sweat chloride (mmol/L)	82.37 ± 15.29	59.68 ± 19.73	<0.001
Glucose tolerance categories:			0.94
NGT n (%)	8 (50.0)	8 (50.0)	
INDET n (%)	2 (12.5)	3 (18.75)	
IGT n (%)	3 (18.75)	2 (12.5)	
CFRD n (%)	3 (18.75)	3 (18.75)	
DC ((pmol·m^−2^·BSA)/(mmol·L^−1^·min^−1^))	2157.0 ± 607.8	1778.4 ± 368.0	0.48
PC (pmol/min/m^2^ BSA)			
ISR_4_	60.9 (41.0–88.5)	61.2 (46.2–84.7)	0.16
ISR_5.5_	123.4 (77.4–235.6)	100.5 (71.0–216.7)	0.50
ISR_8_	340.6 (237.5–533.2)	346.9 (257.2–504.8)	0.09
ISR_11_	619.9 (419.5–907.7)	642.7 (472.0–937.4)	0.91
ISR_15_	970.1 (661.0–1419.8)	1037.1 (805.6–1449.7)	0.11
PC_adj_ ((pmol/L) (mmol/L))	77.6 ± 13.9	98.6 ± 15.2	0.16
Insulin Clearance	1.10 (0.82–1.51)	0.98 (0.88–1.40)	0.26
OGIS	464.1 ± 21.1	403.0 ± 9.8	0.004

**Table 2 jcm-11-04149-t002:** Clinical and metabolic characteristics of the subjects before starting Elexacaftor-Ivacaftor-Tezacaftor and after 12–18 months of treatment. Data are expressed as mean ± SD or median [IQR], unless otherwise specified. Abbreviations: BMI body mass index, FEV1 Forced Expiratory volume in the 1st second, FVC forced vital capacity. NGT normal glucose tolerance, INDET indeterminate glucose tolerance, IGT impaired glucose tolerance, CFRD cystic fibrosis related diabetes, DC derivative control, PC proportional control, ISR insulin secretion rate at 4, 5.5, 8, 11 and 15 mmol/L of plasma glucose, OGIS Oral Glucose Insulin Sensitivity.

Variables	Baseline	After 12−18 months of Exacaftor/Ivacaftor/Tezacaftor	*p*
Gender (M/F) n (%)	4 (80)/1 (20)	−	−
Age (years)	22.0 ± 7.4	23.14 ± 7.50	0.028
Pubertal status			1.00
Pre−pubertal n (%)	0 (0)	0 (0)	
Pubertal n (%)	1 (20)	1 (20)	
Post−pubertal n (%)	4 (80)	4 (80)	
Height (Z−score)	−0.32 ± 1.5	−0.29 ± 1.47	0.48
Weight (Z−score)	−0.67 ± 0.71	−0.43 ± 0.53	0.18
BMI (kg × m^−2^)	19.92 ± 3.0	21.10 ± 2.49	0.07
BMI z−score	−0.90 ± 0.72	−0.44 ± 0.85	0.13
FEV1 (L)	1.67 ± 0.56	2.11 ± 0.57	0.023
FEV1% of predicted	39.0 ± 11.14	49.00 ± 11.27	0.041
FVC (L)	3.57 ± 0.98	4.17 ± 0.95	0.014
FVC% of predicted	72.67 ± 18.18	80.67 ± 10.07	0.287
Sweat chloride (mmol/L)	98.67 ± 15.04	30.33 ± 11.72	0.002
Glucose tolerance categories:			0.07
NGT n (%)	2 (40.0)	2 (40.0)	
INDET n (%)	1 (20)	0 (0)	
IGT n (%)	1 (20)	2 (40)	
CFRD n (%)	1 (20)	1 (20)	
DC ((pmol·m^−2^·BSA) /(mmol·L ^−1^ ·min^−1^))	400.7 ± 169.7	807.8 ± 341.8	0.35
PC (pmol/min/m^2^ BSA)			
ISR_4_	61.7 [45.2−76.4]	66.8 [56.3−89.3]	0.35
ISR_5.5_	111.4 [79.3−167.9]	105.3 [81.6−145.1]	0.89
ISR_8_	258.3 [221.2−475.4]	259.5 [220.3−522.9]	0.69
ISR_11_	523.1 [382.9−844.5]	484.0 [387.1−976.3]	0.50
ISR_15_	882.6 [595.0−1336.5]	804.9 [598.7−1580.9]	0.69
PC_adj_ ((pmol/L)·(mmol/L))	81.6 ± 28.4	88.2 ± 25.3	0.89
Insulin Clearance (L/min)	0.95 [0.83−1.14]	0.92 [0.77−1.15]	0.50
OGIS (µmol·min^−1^·m^−2^ BSA)	449.2 ± 44.4	396.2 ± 25.4	0.14

## Data Availability

The datasets generated analysed during the current study are available from the corresponding author on reasonable request.

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
