# Peer review of "Impact of CFTR Modulators on Beta-Cell Function in Children and Young Adults with Cystic Fibrosis"

_jcm, 2022, doi:10.3390/jcm11144149_

Round 1
Reviewer 1 Report
The manuscript by Piona C. et al., discusses the effect of CFTR modulators, such as: Lumacaftor, Ivacaftor, Elexacaftor and Tezacaftor, used in different combinations on pancreatic beta cell function. The study is well described and holds a clinical relevance in the context of understanding the effects of the above listed drugs used to ameliorate the effects of CF. However, the authors should address the following comments:
1. Introduction, lines 55-6: please specify what is known in the published literature about CFTR modulation and pancreatic beta cell function
2.Results- please add statistical test description and significance (if applicable) to the figure legends/figures
3.Conclusions: Please discuss the possible cause of a significantly increased fasting blood glucose in the group treated with the combination of Lumacaftor and Ivacaftor, as well as decreased HbA1c levels in Ivacaftor, Elexacaftor and Tezacaftor-treated group
4.Conclusions: discussion of what happens with data/conclusions when dissecting the results obtained individually in different glucose tolerance groups is missing- this aspect should also be addressed in the study limitation part
5.The following fragments require style correction: line line 47- "Several evidence", should be "Several reports"; line 243 "in both the", should be "in either of the group"
Author Response
June 24th, 2022
Dr. Vito Terlizzi
Prof. Dr. Stefano Stagi
Guest Editors of the Special Issue "Cystic Fibrosis: Genotype, Clinical Features, and Current
Comorbidities" of the Journal of Clinical Medicine
Object: Revised manuscript submission – manuscript ID jcm-1751349 entitled “IMPACT OF
CFTR MODULATORS ON BETA-CELL FUNCTION IN CHILDREN AND YOUNG ADULTS
WITH CYSTIC FIBROSIS.”
Dear Editors,
Dear Reviewers,
We would like to thank the Editor and the Reviewers for their helpful comments and suggestions on our manuscript. We enclose point by point answers to the reviewers' comments and we modified the manuscript showing, in red and underlined, the revised part.
Please find below our comments and responses to the issues pointed out by the Reviewers.
We appreciate the thoroughness with which the Reviewers regarded our paper and we hope that the article may now be suitable for publication in Journal of Clinical Medicine.
Best regards,
Dr.Antonella Tosco, MD (corresponding author)
Regional Pediatric Cystic Fibrosis Center, Department of Translational Medical Sciences, Section of Pediatrics, Federico II University of Naples, 80131, Naples, Italy.
Contact Phone Number: +39 3343813223
e-mail: [email protected]
Manuscript ID jcm-1751349
Point by Point Replay
Reviewers' Comments to Author:
Reviewer 1
The manuscript by Piona C. et al., discusses the effect of CFTR modulators, such as: Lumacaftor, Ivacaftor, Elexacaftor and Tezacaftor, used in different combinations on pancreatic beta cell function. The study is well described and holds a clinical relevance in the context of understanding the effects of the above listed drugs used to ameliorate the effects of CF.
We thank you for your constructive assessment of our work. We are grateful for the comprehensive revision and recommendation provided that contribute to improve the quality of the manuscript. We addressed all points and included a point-by-point response below.
However, the authors should address the following comments:
- Introduction, lines 55-6: please specify what is known in the published literature about CFTR modulation and pancreatic beta cell function
According to the Reviewer’s suggestion, we better specify what is known in the published literature about CFTR modulation and pancreatic beta cell function (lines 106 – 121). In particular, we more deeply described the studies conducted to date regarding CFTR expression and function in beta-cells addressing also some controversial aspects of this topic.
- Results- please add statistical test description and significance (if applicable) to the figure legends/figures
According to the Reviewer’s suggestion, we add statistical test description and significance to legends of figure 1 and 2 (Lines 589-591 and 607-610).
- Conclusions: Please discuss the possible cause of a significantly increased fasting blood glucose in the group treated with the combination of Lumacaftor and Ivacaftor, as well as decreased HbA1c levels in Ivacaftor, Elexacaftor and Tezacaftor-treated group.
According to the Reviewer’s suggestion, we better discuss both these results in the conclusion section of the manuscript (lines 402-404 and 414).
- Conclusions: discussion of what happens with data/conclusions when dissecting the results obtained individually in different glucose tolerance groups is missing- this aspect should also be addressed in the study limitation part
According to the Reviewer’s suggestion, we add this limitation in the manuscript (lines 417-418).
- The following fragments require style correction: line line 47- "Several evidence", should be "Several reports"; line 243 "in both the", should be "in either of the group"
According to the Reviewer’s suggestion, we corrected the style of these two fragments (line 100 and line 324).

Reviewer 2 Report
This is a prospective study on the effect of 2 different CFTR modulator therapies on glucose metabolism in patients with cystic fibrosis. Changes in insulin secretion (both first and second phase), insulin clearance and insulin sensitivity were evaluated in 16 patients who received Lumacaftor/Ivacaftor and 5 patients who received Elexacaftor-Ivacaftor-Tezacaftor using OGTT measures, before and 12-18 months after initiation of treatment. There was no change in these measures or in the glucose tolerance status of these patients with treatment except for worsening insulin sensitivity in the Lumacaftor/Ivacaftor group. The authors concluded that treatment with CFTR modulators does not improve glucose metabolism in CF patients.
- - The authors mention in the second paragraph of the introduction, that CFTR is expressed in the pancreatic beta cells citing few studies that demonstrated that. However, this is not confirmed, and it is a controversial subject. Hart et al. reported that CFTR deletion from mouse beta-cells did not affect glucose tolerance. They also found minimal CFTR mRNA expression and no detectable CFTR protein in human islet cells (Hart N.J. et al. Cystic fibrosis-related diabetes is caused by islet loss and inflammation. JCI Insight. 2018;3(8)). This controversy should be included as it could help explain why modulator therapy did not affect glucose metabolism.
- -While the Elexacaftor-Ivacaftor-Tezacaftor is considered a highly effective modulator because of the significant improvements seen in lung function, the Lumacaftor/Ivacaftor is not as effective because the improvement in lung function was modest. This should be clarified as they can have different impact on glucose metabolism because of that.
- - It would be helpful to clarify the indications for these 2 different modulator therapies in Italy and why the first group was started on Lumacaftor/Ivacaftor and not Elexacaftor-Ivacaftor-Tezacaftor.
- - Can the authors include the insulin, glucose and c-peptide assays used, and were the assays the same or comparable in the 2 different laboratories used?
- - Can the authors include explanation for the abbreviations ISR4, 5.5, 8, 11 and 15 in the footnotes of the 2 tables.
- - The patients on Elexacaftor-Ivacaftor-Tezacaftor seem to have much lower lung function. Is there an explanation for that? Did they have a more advanced disease?
- - How do the authors explain the decrease in insulin sensitivity seen with Lumacaftor/Ivacaftor?
- - I suggest showing the individual DC data for the 5 patients before and after the Elexacaftor-Ivacaftor-Tezacaftor in a figure, rather than showing the mean in figure 1. Because of the small number of patients, there might be wide variation in response that is better represented if shown individually rather than by the mean.
Author Response
June 24th, 2022
Dr. Vito Terlizzi
Prof. Dr. Stefano Stagi
Guest Editors of the Special Issue "Cystic Fibrosis: Genotype, Clinical Features, and Current
Comorbidities" of the Journal of Clinical Medicine
Object: Revised manuscript submission – manuscript ID jcm-1751349 entitled “IMPACT OF
CFTR MODULATORS ON BETA-CELL FUNCTION IN CHILDREN AND YOUNG ADULTS
WITH CYSTIC FIBROSIS.”
Dear Editors,
Dear Reviewers,
We would like to thank the Editor and the Reviewers for their helpful comments and suggestions on our manuscript. We enclose point by point answers to the reviewers' comments and we modified the manuscript showing, in red and underlined, the revised part.
Please find below our comments and responses to the issues pointed out by the Reviewers.
We appreciate the thoroughness with which the Reviewers regarded our paper and we hope that the article may now be suitable for publication in Journal of Clinical Medicine.
Best regards,
Dr.Antonella Tosco, MD (corresponding author)
Regional Pediatric Cystic Fibrosis Center, Department of Translational Medical Sciences, Section of Pediatrics, Federico II University of Naples, 80131, Naples, Italy.
Contact Phone Number: +39 3343813223
e-mail: [email protected]
Manuscript ID jcm-1751349
Point by Point Replay
Reviewers' Comments to Author:
Reviewer 2
This is a prospective study on the effect of 2 different CFTR modulator therapies on glucose metabolism in patients with cystic fibrosis. Changes in insulin secretion (both first and second phase), insulin clearance and insulin sensitivity were evaluated in 16 patients who received Lumacaftor/Ivacaftor and 5 patients who received Elexacaftor-Ivacaftor-Tezacaftor using OGTT measures, before and 12-18 months after initiation of treatment. There was no change in these measures or in the glucose tolerance status of these patients with treatment except for worsening insulin sensitivity in the Lumacaftor/Ivacaftor group. The authors concluded that treatment with CFTR modulators does not improve glucose metabolism in CF patients.
We thank you for your constructive assessment of our work. We are grateful for the comprehensive revision and recommendation provided that contribute to improve the quality of the manuscript. We addressed all points and included a point-by-point response below.
- The authors mention in the second paragraph of the introduction, that CFTR is expressed in the pancreatic beta cells citing few studies that demonstrated that. However, this is not confirmed, and it is a controversial subject. Hart et al. reported that CFTR deletion from mouse beta-cells did not affect glucose tolerance. They also found minimal CFTR mRNA expression and no detectable CFTR protein in human islet cells (Hart N.J. et al. Cystic fibrosis-related diabetes is caused by islet loss and inflammation. JCI Insight. 2018;3(8)). This controversy should be included as it could help explain why modulator therapy did not affect glucose metabolism.
According to the reviewer’s suggestion the second paragraph of the introduction was changed mentioning that the studies on cystic fibrosis transmembrane conductance regulator protein (CFTR) channel expression in pancreatic beta-cell conducted to date showed conflicting results and adding more information about the study conducted by Hart NJ et al. (Lines 106-121).
- While the Elexacaftor-Ivacaftor-Tezacaftor is considered a highly effective modulator because of the significant improvements seen in lung function, the Lumacaftor/Ivacaftor is not as effective because the improvement in lung function was modest. This should be clarified as they can have different impact on glucose metabolism because of that.
As suggested by the reviewers, it is now known that the efficacy of ETI therapy is superior to that of Lumacaftor / Ivacaftor therapy. We added one sentence in the Introduction (Lines 110-113) and one in the Conclusions (Lines 336-338).
- It would be helpful to clarify the indications for these 2 different modulator therapies in Italy and why the first group was started on Lumacaftor/Ivacaftor and not Elexacaftor-Ivacaftor-Tezacaftor.
In Italy, at the time of enrollment, Lumacaftor/ Ivacaftor was prescribable for patients with homozygous F508del. ETI was indicated for compassionate use for patients over 12 years with advanced lung disease (FEV1 %<40%) with at least one F508del mutation. We added this information in Materials and Methods (Lines 165-166).
-Can the authors include the insulin, glucose and c-peptide assays used, and were the assays the same or comparable in the 2 different laboratories used?
According to the reviewer ‘s suggestion, specific information about the insulin, glucose and c-peptide assays used were added to the material and method section of the paper (lines 193-199).
- Can the authors include explanation for the abbreviations ISR4, 5.5, 8, 11 and 15 in the footnotes of the 2 tables.
According to the Reviewer’s suggestion, we add the requested explanation (lines 575 and 583-584).
- The patients on Elexacaftor-Ivacaftor-Tezacaftor seem to have much lower lung function. Is there an explanation for that? Did they have a more advanced disease?
As the reviewer noted, the lung function of the patients treated with ETI was lower than that of the Lumacaftor / Ivacaftor treated group. At the time of enrollment, in Italy ETI was only indicated for compassionate use, for patients with advanced lung disease and FEV1 below 40%. We added this information in Materials and Methods (Lines 165-166).
- How do the authors explain the decrease in insulin sensitivity seen with Lumacaftor/Ivacaftor?
Explain the reason of the decrease in insulin sensitivity observed in patients treated with Lumacaftor/Ivacaftor is difficult according to current evidence on this topic. Longitudinal data regarding change in insulin sensitivity in patients with CF, and in particular those treated with CFTR modulators, evaluated with methods comparable to the one used in study are not currently available. Indeed, several factors, such as age, clinical course, inflammation as well as possible change in glucose tolerance status during the time could influence insulin sensitivity.
According to our results, we could speculate that this slight decrease demonstrated in patients with Lumacaftor/Ivacaftor is due to physiological compensation mechanisms in response to the slight improvement in PC. Moreover, in lines with this possible explanation and as explained in lines 311-317 and 389-394, we would like to further underline that the vector plots analysis showed that glucose regulation was stable in patients treated with Lumacaftor/Ivacaftor
- I suggest showing the individual DC data for the 5 patients before and after the Elexacaftor-Ivacaftor-Tezacaftor in a figure, rather than showing the mean in figure 1. Because of the small number of patients, there might be wide variation in response that is better represented if shown individually rather than by the mean.
We thank the reviewer for this comment because it gave use the possibility of add some information about derivative control and the approach to its analysis adopted in this study. The first phase of secretion, represented by the derivative control, has a wide variability intrinsic to the insulin secretion physiology: the value of the derivative control has a very wide range and it goes from 0 to over 1200. In our sample, indeed, the variability is accentuated by the small number of subjects. Therefore, to reduce the individual variability and aiming to capture a global signal we decided to look at the data in complexity. We applied this approach to the whole variables analyzed. Moreover, since there is a variation/progression/regression of glycemic status precisely intrinsic to the pathophysiology of CFRD, we believe that global assessment in this case is more suitable to highlight a possible effect of CFTR modulators.

Reviewer 3 Report
Overall, this is an important contribution to the literature, as there as much interest in the effects of the latest CFTR modulator therapies on the pathophysiology of CFRD.
Comments -
Intro and/or discussion -- I would include some mention of the differences between Lum/iva and ETI in terms of CFTR modulation efficacy. ETI is considered a more highly effective modulator, and might be anticipated to exhibit greater changes in glucose metabolism, so it is good that the two groups were analyzed separately. Unfortunately this did result in a rather low N for the ETI group.
Methods - please provide more details regarding the assays used, given the inherent variability that can exist between assays, particular with insulin assays. Because there were 2 institutions participating, were all labs run on the same assay at one institution?
Discussion
Can the authors speculate as to why insulin sensitivity may have decreased in the lum/iva group?
Although the results were not statistically significant, what do the authors make of the differences in the directionality of the change in DC between the two groups? ie - why did it decrease in the LI group and increase in the ETI group?
Author Response
June 24th, 2022
Dr. Vito Terlizzi
Prof. Dr. Stefano Stagi
Guest Editors of the Special Issue "Cystic Fibrosis: Genotype, Clinical Features, and Current
Comorbidities" of the Journal of Clinical Medicine
Object: Revised manuscript submission – manuscript ID jcm-1751349 entitled “IMPACT OF
CFTR MODULATORS ON BETA-CELL FUNCTION IN CHILDREN AND YOUNG ADULTS
WITH CYSTIC FIBROSIS.”
Dear Editors,
Dear Reviewers,
We would like to thank the Editor and the Reviewers for their helpful comments and suggestions on our manuscript. We enclose point by point answers to the reviewers' comments and we modified the manuscript showing, in red and underlined, the revised part.
Please find below our comments and responses to the issues pointed out by the Reviewers.
We appreciate the thoroughness with which the Reviewers regarded our paper and we hope that the article may now be suitable for publication in Journal of Clinical Medicine.
Best regards,
Dr.Antonella Tosco, MD (corresponding author)
Regional Pediatric Cystic Fibrosis Center, Department of Translational Medical Sciences, Section of Pediatrics, Federico II University of Naples, 80131, Naples, Italy.
Contact Phone Number: +39 3343813223
e-mail: [email protected]
Manuscript ID jcm-1751349
Point by Point Replay
Reviewers' Comments to Author:
Reviewer 3
Overall, this is an important contribution to the literature, as there as much interest in the effects of the latest CFTR modulator therapies on the pathophysiology of CFRD.
We thank you for your constructive assessment of our work. We are grateful for the comprehensive revision and recommendation provided that contribute to improve the quality of the manuscript. We addressed all points and included a point-by-point response below.
Comments -
Intro and/or discussion -- I would include some mention of the differences between Lum/iva and ETI in terms of CFTR modulation efficacy. ETI is considered a more highly effective modulator, and might be anticipated to exhibit greater changes in glucose metabolism, so it is good that the two groups were analyzed separately. Unfortunately this did result in a rather low N for the ETI group.
According to the reviewer’s suggestion, we addressed the differences between Lum/iva and ETI in terms of CFTR modulation efficacy in the introduction (110- 113) and in the conclusion (Lines 336-338).
Methods - please provide more details regarding the assays used, given the inherent variability that can exist between assays, particular with insulin assays. Because there were 2 institutions participating, were all labs run on the same assay at one institution?
According to the reviewer ‘s suggestion, specific information about the insulin, glucose and c-peptide assays used were added to the material and method section of the paper (lines 193-199). The intra-assay evaluation, in which was performed the same method on identical samples, in the same laboratory, by the same operator, using the same equipment and in short time intervals (same analytical session), was obtained applying the formula Coefficient of variation (CV)= standard deviation (SD) of the measurements/mean of measurements x 100. The CV of the Verona Central Laboratory of glucose, insulin and c-peptide were 2.3%, 3.4% and 4.4%, respectively. In the Napoli Central Laboratory, the CV, instead, were 2.1%, 4.1% and 5.3%.
Discussion
Can the authors speculate as to why insulin sensitivity may have decreased in the lum/iva group?
Explain the reason of the decrease in insulin sensitivity observed in patients treated with Lumacaftor/Ivacaftor is difficult according to current evidence on this topic. Longitudinal data regarding change in insulin sensitivity in patients with CF, and in particular those treated with CFTR modulators, evaluated with methods comparable to the one used in study are not currently available. Indeed, several factors, such as age, clinical course, inflammation as well as possible change in glucose tolerance status during the time could influence insulin sensitivity.
According to our results, we could speculate that this slight decrease demonstrated in patients with Lumacaftor/Ivacaftor is due to physiological compensation mechanisms in response to the slight improvement in PC. Moreover, in lines with this possible explanation and as explained in lines 311-317 and 389-394, we would like to further underline that the vector plots analysis showed that glucose regulation was stable in patients treated with Lumacaftor/Ivacaftor
Although the results were not statistically significant, what do the authors make of the differences in the directionality of the change in DC between the two groups? ie - why did it decrease in the LI group and increase in the ETI group?
As rightly commented by the Reviewer, no significant changes in DC were found. Thus, the differences in the directionality of the change in DC between the two groups could not be interpreted as indicative of a specific and significative change in this determinant of glucose regulation. Moreover, we would like to underline that vector plot analysis is the only method to deeply evaluate glucose regulation over the time.

Reviewer 4 Report
The manuscript examines beta cell function in CF patients before and after modulators. The findings are consistent with other groups that observed pancreatic function remains impaired in CF patients treated with modulators.
Lines 259-260: It is difficult to extend a pattern to a mechanism. Mechanism implies that there is an experiment with an intervention. This statement needs to be re-worked. Further, it is unclear what the mechanism is. The reviewer suggests changing this statement to “The absence of changes in glucose regulation after treatment suggests that…”
Lines 302-306: The reference does not refer to Cl– secretion. The role of CFTR dysfunction in beta-cells is not as clearly defined as conveyed by the authors. A major reason is that beta cells are already damaged in relevant CF animal models that present with CFRD. There are some functional data for CFTR in subpopulations of beta cells (dois 10.1371/journal.pone.0242749 and 10.1210/clinem/dgz209)
Further English editing is recommended. Specifically, there are many sentence fragments and several paragraphs that consist of 1 sentence.
Deeper modeling was noted as a motivation to perform this study in the discussion. This motivation should be noted in the introduction with a description of the approach for those familiar with CF, but not endocrinologists. There should be a deeper description of the modeling in the methods rather than referring to publications. A more didactic approach could be used for describing findings using these models.
Minor Comments:
Line 23: Period needed in the abstract.
Line 26: In the abstract change the first instance of OGTT to "oral glucose tolerance test (OGTT)"
Line 44: CFRD may be considered a part of CF and not comorbidity by some. The reviewer suggests changing the opening line to “Cystic fibrosis related diabetes (CFRD) affects at least half of adult CF patients.”
Line 258: At first occurrence in the manuscript body change NGT to “normal glucose tolerance (NGT)”
Figure 1: Panel A font is inconsistent with other panels
Author Response
June 24th, 2022
Dr. Vito Terlizzi
Prof. Dr. Stefano Stagi
Guest Editors of the Special Issue "Cystic Fibrosis: Genotype, Clinical Features, and Current
Comorbidities" of the Journal of Clinical Medicine
Object: Revised manuscript submission – manuscript ID jcm-1751349 entitled “IMPACT OF
CFTR MODULATORS ON BETA-CELL FUNCTION IN CHILDREN AND YOUNG ADULTS
WITH CYSTIC FIBROSIS.”
Dear Editors,
Dear Reviewers,
We would like to thank the Editor and the Reviewers for their helpful comments and suggestions on our manuscript. We enclose point by point answers to the reviewers' comments and we modified the manuscript showing, in red and underlined, the revised part.
Please find below our comments and responses to the issues pointed out by the Reviewers.
We appreciate the thoroughness with which the Reviewers regarded our paper and we hope that the article may now be suitable for publication in Journal of Clinical Medicine.
Best regards,
Dr.Antonella Tosco, MD (corresponding author)
Regional Pediatric Cystic Fibrosis Center, Department of Translational Medical Sciences, Section of Pediatrics, Federico II University of Naples, 80131, Naples, Italy.
Contact Phone Number: +39 3343813223
e-mail: [email protected]
Manuscript ID jcm-1751349
Point by Point Replay
Reviewers' Comments to Author:
Reviewer 4
The manuscript examines beta cell function in CF patients before and after modulators. The findings are consistent with other groups that observed pancreatic function remains impaired in CF patients treated with modulators.
We thank you for your constructive assessment of our work. We are grateful for the comprehensive revision and recommendation provided that contribute to improve the quality of the manuscript. We addressed all points and included a point-by-point response below.
Lines 259-260: It is difficult to extend a pattern to a mechanism. Mechanism implies that there is an experiment with an intervention. This statement needs to be re-worked. Further, it is unclear what the mechanism is. The reviewer suggests changing this statement to “The absence of changes in glucose regulation after treatment suggests that…”
According to the Reviewer’s suggestion, the statement was changed and simplified trimming the part regarding mechanistic explanation (Lines 316).
Lines 302-306: The reference does not refer to Cl– secretion. The role of CFTR dysfunction in beta-cells is not as clearly defined as conveyed by the authors. A major reason is that beta cells are already damaged in relevant CF animal models that present with CFRD. There are some functional data for CFTR in subpopulations of beta cells (dois 10.1371/journal.pone.0242749 and 10.1210/clinem/dgz209)
According to the reviewer’s suggestion, we more deeply analyzed and discussed the role of CFTR dysfunction in beta-cells focusing also on the role of chloride transporters and channels, including CFTR, in β-cell physiology (Lines 339-354)
Further English editing is recommended. Specifically, there are many sentence fragments and several paragraphs that consist of 1 sentence.
According to the Reviewer’s suggestion, a mother tongue English copy editor edited the revised version of the manuscript.
Deeper modeling was noted as a motivation to perform this study in the discussion. This motivation should be noted in the introduction with a description of the approach for those familiar with CF, but not endocrinologists.
According to the reviewer’s suggestion, we better describe the modelling approach in the manuscript (Lines 142-144 and material and method section).
There should be a deeper description of the modeling in the methods rather than referring to publications.
According to the reviewer’s suggestion, we deeper describe the modeling in the methods (Lines 225- 265).
A more didactic approach could be used for describing findings using these models.
According to the reviewer’s suggestion, we modify several sections of the manuscript adding more details about the modeling approach and further describing our results in order to make both the methods and the finding more easily understandable (Lines 142-144 and material and method section).
Minor Comments:
Line 23: Period needed in the abstract.
Line 26: In the abstract change the first instance of OGTT to "oral glucose tolerance test (OGTT)"
Line 44: CFRD may be considered a part of CF and not comorbidity by some. The reviewer suggests changing the opening line to “Cystic fibrosis related diabetes (CFRD) affects at least half of adult CF patients.”
Line 258: At first occurrence in the manuscript body change NGT to “normal glucose tolerance (NGT)”
Figure 1: Panel A font is inconsistent with other panels
We modified the manuscript according to these minor comments expressed by the reviewer.

Round 2
Reviewer 1 Report
The authors addressed the comments.